# A New Modular Petri Net for Modeling Large Discrete-Event Systems: A Proposal Based on the Literature Study

**Reggie Davidrajuh** 

Electrical & Computer Engineering, University of Stavanger, 4036 Stavanger, Norway;
Reggie.Davidrajuh@uis.no; Tel.: +47-518-310-51

**Abstract:** Petri net is a highly useful tool for modeling of discrete-event systems. However, Petri net models of real-life systems are enormous, and their state-spaces are usually of infinite size. Thus, performing analysis on the model becomes difficult. Hence, slicing of Petri Net is suggested to reduce the size of the Petri nets. However, the existing slicing algorithms are ineffective for real-world systems. Therefore, there is a need for alternative methodologies for slicing that are effective for Petri net models of large real-life systems. This paper proposes a new Modular Petri Net as a solution. In modular Petri net, large Petri net models are decomposed into modules. These modules are compact, and the state spaces of these modules are also compact enough to be exhaustively analyzed. The research contributions of this paper are the following: Firstly, an exhaustive literature study is done on Modular Petri Nets. Secondly, from the conclusions drawn from the literature study, a new Petri net is proposed that supports module composition with clearly defined syntax. Thirdly, the new Petri net is implemented in the software GPenSIM, which is crucial so that real-life discrete-event systems could be modeled, analyzed, and performance-optimized with GPenSIM.

**Keywords:** Modular Petri Net; discrete-event simulation; GPenSIM

## 1. Introduction

Petri nets have been used for modeling, simulation, performance analysis, and control of discrete-event systems. The wide acceptance of Petri nets is due to its well-known properties such as graphical (visual) representation that closely resemble real-life objects and formal & well-defined semantics [1–3]. Petri net has a strong yet simple mathematical background which is limited to linear algebraic techniques and graph theorems. The background mathematics enable thorough system analysis (such as state-space analysis, performance bottlenecks, and deadlock avoidance) [4]. A large number of Petri net software tools are also available, some of them for specific purposes and some for general simulations (e.g., CPN and GPenSIM) [5,6]. There are several Petri net extensions also in use. Some of these extensions increase the modeling power while preserving its analytical power (e.g., Colored Petri Nets). While some other extensions make a trade-off (e.g., state machines and marked graphs increase the analytical power while slightly reducing the model power) [7]. This paper is on modular Petri nets, for partitioning Petri net models into modules for ease of model building and ease of analysis.

In this paper: Section 2 presents a thorough literature review on modular Petri nets. Section 3 explores the topic of independent module development. The ideas gathered from the first two sections are used in Sections 4 and 5 for the design of modular Petri nets. Section 6 specifies the new Modular

Petri nets, and the formal definitions are given in Section 7. Section 8 presents an application example. Finally, Section 9 is for discussion.

## 2. Literature Review on Modular Petri Nets

The classic Petri net (*aka* P/T Petri net) does not provide any support for modularization. Modular model building with Petri nets has a short history, starting with works in the late 1990s. This section presents a literature review on this topic. The conclusions drawn from the literature study is summarized at the end of this section. These conclusions are the basis for the new design of Petri Net modules presented in this paper.

### 2.1. The Problem with Petri Net Models

There are some problems associated with modeling real-life discrete-event systems with Petri nets:

- Petri net models of real-life systems are huge: Even for simple systems, the Petri net models of these are not small or compact [8].
- Slowness in simulation: During simulations, the tokens in a Petri net have to go through every transition and place on their path. Also, the transitions have to be checked for their enabledness and the other firing conditions from the environment. This makes the simulations run slowly. Also, the enormous sizes of Petri nets contribute to slowness in simulation.
- Difficulty in analyzing the model: Again, due to the huge size of the model, analyzing the model for its structural and behavioral properties become a time-consuming task.
- "State Explosion": The most important and useful property of Petri nets is their explicit state space. The state space is automatically generated, showing every possible state that can be eventually reached from an initial state. However, for real-life systems, the state space is huge if not of infinite size. Drawing any conclusions (e.g., model checking) from the huge or infinite state space is often difficult if it is possible at all [9–11].

Literature study provides some slicing algorithms to reduce the size of Petri nets as well as the state space. However, these slicing algorithms though works on small hypothetical example, have little or no effect on real-life discrete-event systems [12].

### 2.2. Literature Review on Modular Petri Nets

In the following subsections, a thorough literature study is done to find whether modules and modular Petri nets can become a solution to the problems listed above.

2.2.1. First-Generation Works: Ease of Modeling

Refs. [13,14] provide, already in the 1990s, a powerful technique for compression of Petri net modules. According to the reduction theorem by [13,14] , if a module is an event graph, and it has transitions as input and output ports, then the module can be compressed into a much smaller module. Refs. [13,14] prove that an event graph that possesses only transitions as input and output ports, then it can be represented by a compact module in which the all internal transitions are removed, and also some internal places are removed. Hence, the whole model becomes modules that are compact and connected together by a few buffering places (since the input and output ports are transitions, the connection between the modules must be places). Also, by the reduction theorem, the liveness and boundedness properties of the original modules are preserved.

Ref. [15] is one of the early works on modular Petri nets. Ref. [15] is concerned about a particular environment, which is the modeling and simulation of interfacing techniques in circuit boards. The paper proposed interfacing at every module-level rather than keeping interfacing techniques in one specific module, which was the norm at that time. The paper state that keeping all the interfacing in one module makes communication between the modules unnecessarily complicated. This paper also proposed designing modules with two-level, the lower-level is for transmission and synchronization

of signals, and the higher-level for communication of messages, resulting in a new communicating Petri net model.

Refs. [16,17] introduced "Object-oriented Petri Nets", to reap the benefits of object-oriented programming also in the modeling of discrete-event systems. These works proposed an object-oriented model building approach in which the generic Petri net modules are declared as classes. Then, from the classes, the specific instances of modules are developed for modeling specific problems.

Ref. [18] proposed a dual approach for dealing with modular systems. This paper proposed a strategy for identifying specific users and modules that only capture the logic interested to those particular users. In other words, the proposal was for a partition of a large model into separate modules that consist of model logic that will be of interest to those in specific interest areas.

Ref. [19] took flexible manufacturing systems as systems with specific subsystems (called subnets). The subnets are identified as transportation of raw material & resources, machining subnets, and finishing subnets.

The contributions of the first generation works are summarized in Table 1.

**Table 1.** First-generation works on ease of modeling.

| Work | Topics |
|------|--------|
| Savi & Xie (1992) [13] and Claver et al. (1991) [14] | Module compression. |
| De & Lin (1994) [15] | Clear-cut interfacing. |
| Wang (1996) [16] and Wang & Wu (1998) [17] | Object-oriented Petri Nets. |
| Lee et al. (1998) [18] | Decomposing a Petri net into modules based on functionality. |
| Xue et al. (1998) [19] | Flexible manufacturing systems as systems with specific subsystems. |

### 2.2.2. Second-Generation Works: Analysis

The first-generation works given in the previous sub-section are on reaping the benefits of the modular model building also with Petri nets. The second-generation works focused on easing the *analysis* of huge and complex Petri nets.

Ref. [20] presented a modularization of Petri nets using *fusion places* and *fusion transitions*. Fusion places and fusion transitions are special types of places and transitions, respectively. These places and transitions are only to partition a Petri net model into modules and analyze them individually, due to the firings of the local (members of the module) transitions. For example, the state spaces of the individual modules can be obtained by the firing of the local transitions, starting with the initial markings on the local places. Then, from the individual state spaces, the overall state space of the model can be obtained by putting together the individual state spaces along with the additional state spaces (known as the "synchronization graph"). Synchronization graph connects the individual state spaces by the firing of the fusion transitions. The novelty of Ref. [20] is that the authors prove that the modularization preserves the main properties of the model (e.g., the place invariants) while removing the need for generating the overall state space which usually suffers state explosion. Ref. [20] also prove that the state space built by the modular approach is much smaller than the state space obtained from the holistic model.

Though fusion places and fusion transitions seem very useful for modular model building, they are against the fundamental concept behind modularization, namely "data hiding"; see the discussion in Section 3.

Ref. [21] focused on reusable generic modules. This work believes that manufacturing systems consist of specific building blocks such as production line, assembly, disassembly, and parallel machining elements. Once these blocks are developed as generic modules, then by customizing these blocks to suit any specific needs, a model of the system could be built and analyzed. Thus, Ref. [21] reinforce the classical benefits of modularization such as speed and easiness of

modeling, and easy adjustment to suit specific needs, as well as analysis. The authors earlier works, such as Ref. [22] also focused on customizing generic modules with fuzzy logic to confront uncertainty in the modeling process.

Ref. [23] proposed "reconfigurable modules" to tackle uncertainties associated with models. This work states that reconfigurable modules support model development, design variations, and cooperative model development. By reconfiguration, this work classifies the uncertainties into two groups variations and ambiguity. Variation in a process means, for example, an operation may take two to three minutes. Ambiguity in a process means, for example, the operation may happen or not depending on certain conditions. Ref. [23] used stochastic modules for tackling variations and fuzzy logic for ambiguity, resulting in a new type of Petri net called a Fuzzy Colored Petri Net with Stochastic time delay (FCPN-std).

Ref. [24] discusses the managerial implications of modularization. This work discusses some issues in modularization, such as the use of "blocks" (modules) for easing the design process and the possibility of reusing the blocks. Also discussed is the issue of redesigning the model by trying out different combinations of modular blocks.

The contributions of the second generation works are summarized in Table 2.

**Table 2.** Second-generation works on analysis of modular Petri Nets.

| Work | Topics |
| --- | --- |
| Christensen & Petrucci (2000) [20] | State space analysis. |
| Tsinarakis et al. (2005) [21]; Tsourveloudis et al. (2000) [22] | Reusable module for ease of analysis. |
| Lee & Banerjee (2009) [23] | Reconfigurable modules to tackle uncertainties associated with models. |
| Latorre-Biel et al. (2017) [24] | Managerial implications of modularization. |

### 2.2.3. Third-Generation Works: Applications & Tools

Ref. [25] presents a tool known as "Exhost-PIPE," for modular timed and colored Petri nets. With the tool, the work shows how a multi-agent environment (e.g., a swarm robot or an aircraft crew) can be modeled and simulated. Ref. [26] presents a modular Petri net model for modeling and simulation of molecular networks. In this work, proteins are represented as Petri net modules. Each module has an interface to access publicly available information about the intra-molecular changes; thus, the modules can update themselves independently. This work presents the design of the interface, the formalized language for modular communication, and the Petri Net model of the molecular network.

A modular Petri net model of the "Spanish National Health System" is described in [27,28]. Refs. [27,28] show the largeness and complexity of the Spanish Health System. These works show that without a modular approach, it would not be able to model and analyze such a large and complex system. In the modular model, each module is independently modeled, keeping the state-machine Petri net as the backbone for modeling the medical protocols. The modules can load the medical resources themselves. Ref. [29] presents a modular p-timed (timing associated with places) Petri net model for analyzing traffic signal control of a network of intersections. This work also shows a light-weight approach for model checking with linear time logic (LTL) based specifications.

Ref. [30] tries to model non-linear process planning (NLPP) in manufacturing systems with a modular Petri net known as Object Observable Petri Net (OOPN). The approach presented in this paper uses three steps. In the first step, the system resources are grouped into two groups: (1) processing resources (e.g., machine tools); and (2) part-flow resources (e.g., conveyor belts and buffers). It is an assumption in the approach that any machining activity uses at least one system resources. In the second step, the model is divided into modules, each module composed of resources with limited capacity. In the final step, each module is converted into a resource operation template

(a Petri net module) adhering to the resource constraints. Ref. [30] uses transitions as the input and output ports for communication between the modules, and thus the communication of a module with the outside world is streamlined through the input and output ports. Though the approach presented in this work is straightforward, and the resulting modules are simple and elegant, the overall model becomes huge. It is also not clear whether a tool (software) is available that can automatically perform the steps involved, or even development of such as a tool is feasible. Without a software tool, it will be impossible to model, even the simplest manufacturing system, with the approach proposed in this paper.

Ref. [31] shows a modular Petri net based approach for detection and elimination of redundancy in virtual enterprises. In the modular Petri net, each module represents one participating enterprise. The participating enterprises on the upstream are raw-material suppliers, part-suppliers, and transporting agents. On the downstream, distributors and sales agents are the participating enterprises. It is noteworthy that all these Petri net modules are event graphs. An event graph is a P/T Petri net in which each place has precisely one input transition and one output transition. Also, the interfaces of the modules are input and output ports that are transitions.

The approach by [31] is elegant as it introduces a clear-cut input and output ports that are transitions. The approach also proposes the use of colored Petri nets. Otherwise, it will be impossible to route a token from a buffering place to the correct module should more than one module is output to that buffering place. Also, forcing all the modules to be devised as event graphs can put a lot of strain on the modeler. This is because "the choice" cannot be modeled in an event graph, and a place gathering tokens from different transitions is also not possible.

Ref. [32] presents a framework for performance evaluation of the intermodal transportation chain in Freight Transport Terminals. The framework is based on a modular timed Petri Nets. In this timed Petri net, places represent resources, capacities and conditions, whereas transitions represent activities such as inputs and flows into the terminal. Finally, tokens represent intermodal transport units. This work uses Generalized Mutual Exclusion Constraints (GMECs, [33]) for realizing the control elements, and the software HYPEN [34] for simulation. Ref. [35] models and analyzes Web service composition. The reason for the analysis is to guarantee the timely completion of the web service. Hence, temporal constraints are emphasized in this work. The problem of state-space explosion is also addressed in this work, albeit not in a transparent manner. This work claims that the model is modular by showing some modules. However, the issue of modularity is not discussed in detail.

The contributions of the third generation works are summarized in Table 3.

**Table 3.** Third-generation works on tools and application of modular Petri nets.

| Work | Topics |
| --- | --- |
| Bonnet-Torres et al. (2006) [25] | Tool: Exhost-PIPE Application: Modeling Multi-Agent environment. |
| Blatke et al. (2011) [26] | Tool: Formal language for modular communication Application: Modeling Molecular Networks. |
| Mahulea et al. (2012) [27] Mahulea et al. (2018) [28] | Application: Modeling Spanish National Health System. |
| Du et al. (2013) [35] | Application: Web Service Composition |
| Dotoli et al. (2016) [32] | Application: Evaluation of Intermodal Freight Transport Terminals |
| Dos & Vrancken (2012) [29] | Tool: Modular Place-Timed Petri net Application: Traffic signal control of network of intersections. |
| Slota et al. (2016) [30] | Tool: Object Observable Petri Net Application: Modeling non-linear process planning in manufacturing systems. |
| Davidrajuh (2013) [31] | Tool: GPenSIM (earlier version)—modules with clear-cut input and output ports Application: Elimination of redundancy in virtual enterprises. |

2.2.4. Fourth-Generation Works: Independent Modules for Modeling Smart Manufacturing

In the era of Smart manufacturing and Industry 4.0, manufacturing systems are composed of interoperable intelligent systems. These intelligent systems are independent and exchange a great amount of data in real-time with their counterparts that are located in geographically separated areas. Finally, smart manufacturing happens via the events that are triggered by networked sensors [36–38].

Ref. [39] is a recent work that develops a modular Petri net for modeling "the availability of risks of IT threats" in Smart Factory Networks (SFN). First, the model is divided into two blocks, one for the information and control network (IN block), and the other for production network (PN block).

The IN block is hierarchical, consisting of three layers. A server is placed on the top layer, which is connected to many IT-service nodes in the middle layer. Each of the IT-service nodes is connected with several machine-control nodes in the bottom layer. These machine-control nodes are the ones that directly interact with the nodes in the PM block. The IN block is hierarchical in a sense each node is connected with several nodes in the lower layer (1:n connection). Whereas a node is connected with only one node in the layer above, forming a tree-like structure. All the nodes in the IN layer (called information component—IC) are the same, simple, and generic Petri net modules.

The PN block consists of nodes (called production machine components—PM) that are connected in a way to represent the logical production and flow of manufactured items. The PM components are also simple and generic Petri net modules. This means, the whole model can be developed with just two simple Petri net modules: one IC module for the nodes in the IN block, and PM module for nodes in the PN block.

Perhaps the approach may solve the prescribed problem (namely, "modeling the availability of risks of IT threats in Smart Factory Networks"). However, the resulting model will be huge. This is because two types of simple generic Peti net modules are repeatedly used to compose any eventual functionality. Besides, the approach uses many Petri net extensions such as inhibitor arcs, reset arcs, and testing arcs. Though the use of these arcs paves a compact module, it prohibits the use of the readily available techniques and algorithms (e.g., for reachability tree). Unique algorithms have to be developed for the analysis of models developed by this approach. However, the availability of such special algorithms and the need for it is not discussed in [39]. The usefulness of this paper is the introduction of a separate hierarchical block (the IN block) that function as the inter-modular connector of the modules in the PM block.

*2.3. Conclusions Drawn from the Literature Study*

The following conclusions can be drawn from the literature study on the works on modular Petri nets. These conclusions will be used in the latter sections on the design of a new modular Petri Net.

- Advantages of modularizing: Literature study reveals that modularization is to reap the benefits such as flexibility (ability to add or change functionality), comprehensibility (readability of the models), reduction in the development time, and robustness (less prone to error).
- On the modularity of systems: Petri net models of real-life discrete-event systems are large and complex. However, these large and complex systems can be modeled as modular models.
- On the scope of a module: A module can trap a specific type of model logic for particular users.
- On interfacing the modules: Old fashioned monolithic pathways (based on master or supervisor) interfacing is not appropriate for inter-modular communication. Interfacing must be at every module level, making the module independently react with the rest of the system and the environment.
- On the design of modules: Timed P/T Petri Net can serve as the skeleton of the module. Timed colored Petri nets are for embedding more detailed data on tokens.
- Attacking ambiguities in the model: Some of the ambiguities can be realized as logical conditions, and can be kept away from the Petri net model (e.g., as logic conditions in processor files).
- On the synchronization of modules [20,30,40–42]:

- Fusion places: The fusion places are for modeling convenience as they are aliases for a place; the fusion places are to eliminate arcs crisscrossing the model. If the fusion places are put in different modules, since they represent the same place, these will be used to synchronize the modules.

- Substitution transition: A substitution transition is for information hiding; a substitution transition represents a complete Petri net module consisting of many places and transitions. Thereby, a substitution transition hides the lower-level details of a module on a higher-level (overall) model.

- Fusion Transitions: A fusion transition (some times referred to as "shared transition") is to allow synchronization of Petri net modules. The shared transitions reside in different modules, but they represent the same transition. Thus, the shared transitions synchronize the modules.

- Communication of a module with the outside world must be streamlined through input and output ports; transitions can be used as ports.

## 3. Towards the Design of a New Modular Petri Net

Table 4 summarizes the history of research and development on modular Petri nets. It started with ease of modeling with modules and then advanced to the analysis of large Petri nets with modular nets. Then, the tools were made available, and some applications began to appear. Finally, the modular approach is tried for reducing the complexities in cyber-physical systems in Industry 4.0.

**Table 4.** The literature on Modular Petri Nets.

| Generation | Topics |
| --- | --- |
| First Generation | **Ease** of modeling. |
| Second Generation | **Analysis** of large Petri Nets. |
| Third Generation | **Tools** for modeling modular Petri nets, and **applications**. |
| Fourth Generation | **Modules:** Modeling large discrete-event systems with modules. |

This paper is to design a modular Petri net belonging to the fourth generation, that is capable of:

- Independent development of modules and analysis (to reduce the complexity of development and analysis of the overall model).
- The modules must be capable of running independently, presumably on different processors (CPU, to reduce the computation time).

*Independent Development of Modules and Analysis*

Let us study the modular approach realized with fusion places and fusion transitions in Ref. [20]. To understand how the methodology works, let us consider the example of a resource allocation system (RAS) given in ref. [20] and also shown in Figure 1.

As the example of fusion places based modular model building, the RAS is remodelled into a two-modules based modular system shown Figure 2.

In the modular RAS shown in Figure 2, the two modules A and B are synchronized by the fusion places $R_X$ and $R_Y$. By the definition of fusion places, the places $R_X$ in the different modules is the same. If any changes happen to a place $R_X$ in one module, the other $R_X$ in the other module will be also affected. This kind of sharing of local members of modules between the modules hampers independent module development.

Let us assume that independent groups develop these two modules. However, due to the sharing $R_X$ and $R_Y$, the developers of module A should always be aware of $R_X$ and $R_Y$ in module B, making the development less independent. The exposure of internal details to the outside world is also against the concept of "data hiding", which is an important concept in modularity.

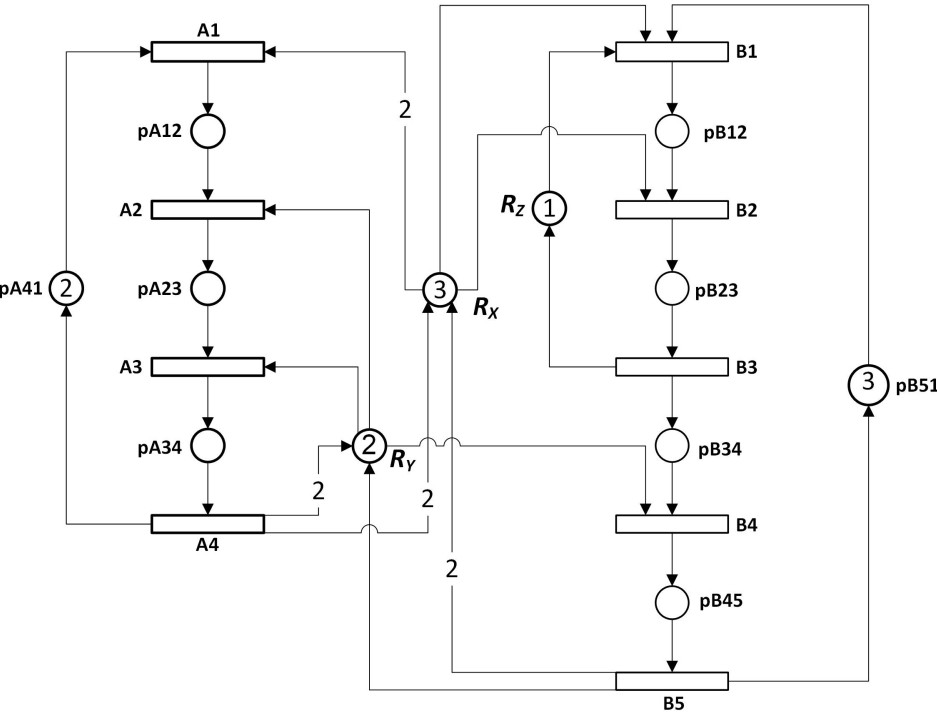

**Figure 1.** Resource Allocation System [20].

Modular model development using fusion transition works very similar to fusion places, and the only difference is that the (fusion) transitions are shared rather than the places. Hence, here too, the modular model building is prone to internal data exposure, hindering data hiding and independent model development.

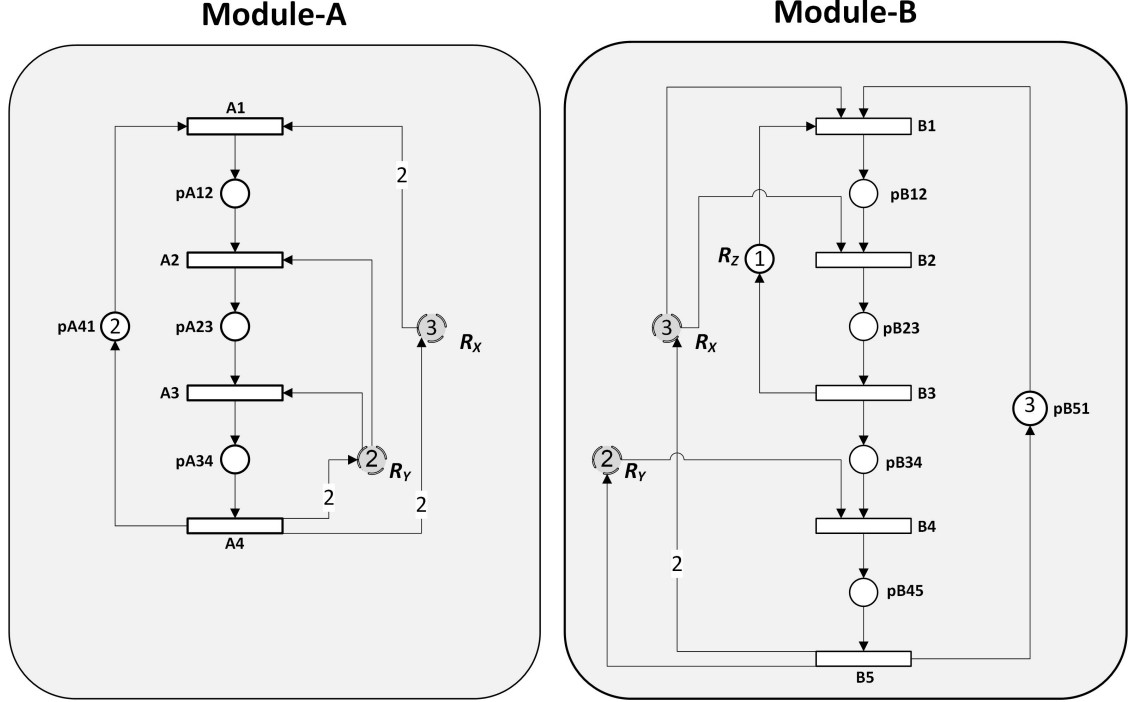

**Figure 2.** Modularization using fusion places [20].

Can the modules that are shown in Figure 2 be improved to become more "modular," following the better practices of module making?

The subsequent sections present the complete design details of a new modular Petri net. As a quick introduction, Figure 3 shows a modular version; in this modular model, RAS is composed of two modules, module-A and module-B, and the **Inter-Modular Connector (IMC)** that consists of just the three places representing the resources $R_X$, $R_Y$, and $R_Z$. The modular model supports data hiding, clear-cut interfacing, and suited for parallel execution:

- Interface to the module: Module-A possesses input ports (A1, tD1, and tD2) and output ports (A4) that function as the input and output interface of the module. The input and out ports have global visibility and can be accessed like global variables.
- Data hiding: Module-A also possesses local members (transitions A2 and A3, and places pA12, pA23, pA34, pD1, and pD2) that have local (modular) visibility, thus can not be seen or accessed outside the module.
- Independent module development: As seen in Figure 3, module-A can be independently developed, with two drivers replacing the places $R_X$ and $R_Y$, and two stubs also replacing the places $R_X$ and $R_Y$.

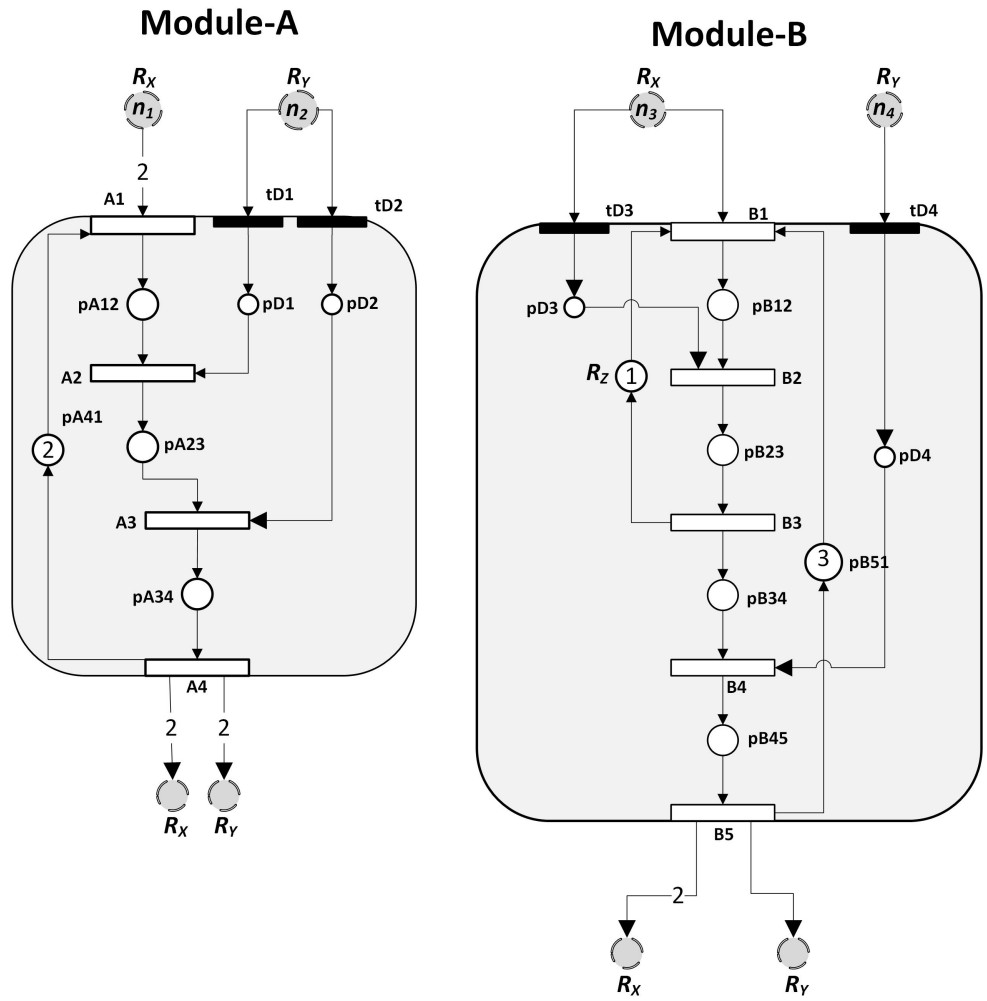

**Figure 3.** Modular Petri net model of RAS: A proposal.

## 4. Design of Modular Petri Nets

The previous section presented some conclusions drawn from the literature study. Partly based on these conclusions, and with the addition of new (modern) ideas, this section presents the unique design of a new Modular Petri Net. At this juncture, it must be emphasized that the approach for modular

Petri net given in this and the following sections are specially designed for GPenSIM implementation. The theory given in this paper and its implementation in the software GPenSIM grew together, and hence the use of GPenSIM terminology (see [43]) in this paper is unavoidable.

### 4.1. Transitions and Their Visibility

There are two key issues behind the design of a modular Petri net. The two key issues are (1) Activity-orientedness and (2) Visibility. These two key issues are also discussed in the book on GPenSIM [43]. The key issues are:

1.  Activity-orientedness: When modeling a discrete-event system, transitions are the primary focus.
2.  Visibility: Transitions possess different visibility, such as global, modular, and private visibility.

Transitions represent the activities of discrete-event systems, whereas places represent the passive elements (e.g., buffers). In a discrete-event system, if there is no activity happening now or in future, then the system is dead. Thus, activities are the heartbeat of discrete-event systems. The places are just drawn along with the transitions. Hence, the transitions representing activities take the central place in the design of modular Petri nets, as well as in GPenSIM.

### 4.2. Visibility of Transitions in a Monolithic Petri Net

In a monolithic (non-modular) Petri net, all transitions have two types of visibility: (1) **global visibility**, and (2) **private visibility**.

#### 4.2.1. Global Visibility

All the transitions in a monolithic Petri net have global visibility. A transition that has global visibility is accessible in the *common* processor files COMMON_PRE and COMMON_POST. Whenever a transition with global visibility becomes enabled, the compiler will automatically check whether there are any pre-conditions in pre-processor file COMMON_PRE the transition has to satisfy before starting to fire. If the transition starts firing, when it completes firing, the post-processor COMMON_POST will be checked for any post-firing actions to be executed.

#### 4.2.2. Private Visibility

Every transition in a monolithic (also in modular Petri net), has private visibility too. Any transition can have its *own* processor files, and in this file the transition is accessible, giving the private visibility. (Transitions in modular Petri net have one more visibility, known as the modular (or local) visibility, which is discussed later in Section 5.2).

As an example: Figure 4 shows a simple monolithic Petri net in which all the three transitions t1, t2, and t3 have global visibility and thus are accessible in the common processor files COMMON_PRE and COMMON_POST. If there are exists processor files t1_pre and t1_post (for the exclusive use of t1), t2_pre and t2_post (for the exclusive use of t2), and so on, then t1 is accessible in t1_pre and t1_post files too, giving private visibility (resp. t2 in t2_pre and t2_post, and t3 in t3_pre and t3_post).

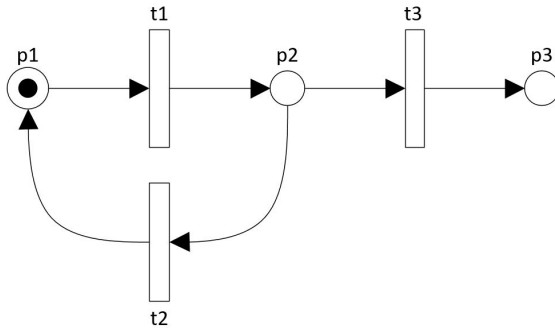

**Figure 4.** A Monolithic P-T Petri Net.

## 5. Composition of Modular Petri Nets

A modular Petri net consists of zero or more Petri Modules. The symbol Φ represents a Petri module. Zero or more Inter-Modular Connectors connects these Petri modules. The symbol Ψ represents an inter-modular connector.

Figure 5 shows a modular Petri net with two Petri modules "Alfa" and "Beta", and two inter-modular connectors (IMC) "Gamma" and "Delta".

What are IMCs? When a modular model is developed, it happens that there exist one or more elements that cannot be included in any of the modules. The reason can be that the model logic of the modules excludes the inclusion, or simply, the element is an inter-module connector. For simplicity, these "leftover" elements can be grouped into a segment (or segments) and be called an IMC (IMCs).

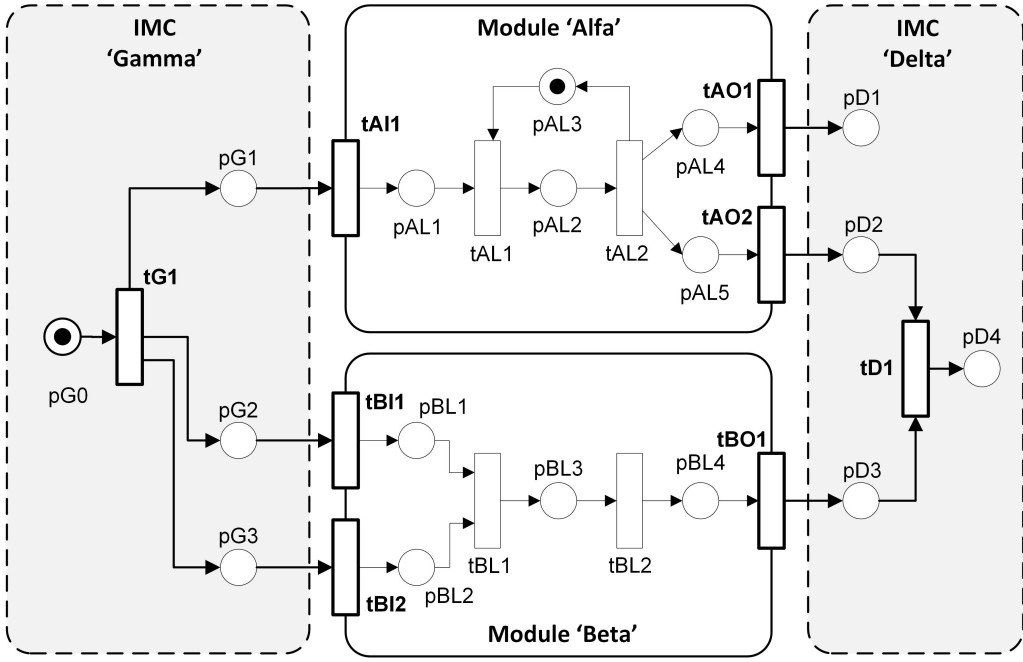

**Figure 5.** A Modular Petri Net with two Modules and two IMCs.

### 5.1. Transitions in Modular Petri Nets

There are four types of transitions in modular Petri nets (see Figure 5):

1. Input Ports: The transitions that function as the input ports of the modules. e.g., tA1 is the input port of Alfa. tBI1 and tBI2 are the input ports of Beta. Thus,
   $$T_{IP\Phi_{Alfa}} = \{tAI1\}$$

$T_{IP\Phi_{Beta}} = \{tBI1, tBI2\}$

$T_{IP} = T_{IP\Phi_{Alfa}} \cup T_{IP\Phi_{Beta}}$

2. Local transitions: The transitions that are internal members (not input or output ports) of modules. e.g., tAL1 and tAL2, and tBL1 and tBL2 are the local transitions of Alfa and Beta, respectively.

$T_{L\Phi_{Alfa}} = \{tAL1, tAL2\}$

$T_{L\Phi_{Beta}} = \{tBL1, tBL2\}$

$T_L = T_{L\Phi_{Alfa}} \cup T_{L\Phi_{Beta}}$

3. Output Ports: The transitions that function as the output ports of the modules. e.g., tAO1 and tAO2 are the output ports of Alfa. tBO1 is the output port of Beta.

$T_{OP\Phi_{Alfa}} = \{tAO1, tAO2\}$

$T_{OP\Phi_{Beta}} = \{tBO1\}$

$T_{OP} = T_{OP\Phi_{Alfa}} \cup T_{OP\Phi_{Beta}}$

4. Inter-Modular transitions: The transitions that are members of Inter-modular connectors. e.g., tG1 in Gamma, and tD1 in Delta.

$T_{IM\Psi_{Gamma}} = \{tG1\}$

$T_{IM\Psi_{Delta}} = \{tD1\}$

$T_{IM} = T_{IM\Psi_{Gamma}} \cup T_{IM\Psi_{Detla}}$ $T_{\Phi_{Alfa}} = T_{IP\Phi_{Alfa}} \cup T_{L\Phi_{Alfa}} \cup T_{OP\Phi_{Alfa}}$ (all the transitions of Alfa)

$T_{\Phi_{Beta}} = T_{IP\Phi_{Beta}} \cup T_{L\Phi_{Beta}} \cup T_{OP\Phi_{Beta}}$ (all the transitions of Beta)

$T = T_{IP} \cup T_L \cup T_{OP} \cup T_{IM}$ (the set of all transitions in the Petri net)

Also, $T = T_{\Phi_{Alfa}} \cup T_{\Phi_{Beta}} \cup T_{IM}$ (the set of all transitions in the Petri net)

There are two types of places in modular Petri nets (see Figure 5):

1. Local places: The places that are local to modules. e.g., pAL1 to pAL5 in Alfa, and pBL1 to pBL4 in Beta.

$P_{L\Phi_{Alfa}} = \{pAL1, \ldots, pAL5\}$ (local places of Alfa)

$P_{L\Phi_{Beta}} = \{pBL1, \ldots, pBL4\}$ (local places of Beta)

$P_L = P_{L\Phi_{Alfa}} \cup P_{L\Phi_{Beta}}$ (local places of all the modules)

2. Inter-Modular places ($P_{IM}$): The places that are members of IMCs. e.g., pG1 to pG3 in Gamma, and pD1 to pD4 in Delta.

$P_{IM\Psi_{Gamma}} = \{pG1, \ldots, pG3\}$ (IM places of Gamma)

$P_{IM\Psi_{Delta}} = \{pD1, \ldots, pD4\}$ (IM places of Delta)

$P_{IM} = P_{IM\Psi_{Gamma}} \cup P_{IM\Psi_{Detla}}$ (IM places of all IMCs) $P = P_L \cup P_{IM}$ (set of all the places in the Petri net)

*5.2. Visibility of Transitions in a Modular Petri Net*

Transitions in modular Petri nets have three different visibility, such as global visibility, local visibility, and private visibility:

1. Inter-modular transitions have global visibility: All the transitions of the inter-modular connectors ($\forall t \in T_{IM}$) have global visibility, thus are accessible in COMMON_PRE and COMMON_POST.

2. Input and output ports have global visibility: All the transitions that are input or output ports of modules ($\forall t \in (T_{IP} \cup T_{OP})$) also have global visibility, thus are accessible in COMMON_PRE and COMMON_POST.

3. Local transitions have modular visibility: Transitions that are local members of modules ($\forall t \in T_L$) have local visibility as they are accessible only in their modular processors MOD_PRE & MOD_POST. For example, transitions tAL1 and tAL2 (tBL1 and tBL2) are local members of the modules Alfa (resp. Beta), and hence are accessible only in their modular processors MOD_Alfa_PRE, MOD_Alfa_POST (resp. MOD_Beta_PRE, MOD_Beta_POST).

   However, these local transitions are not accessible in COMMON_PRE and COMMON_POST, as these transitions do not possess global visibility.

4. Input and output ports have modular visibility too: Transitions that are input or output ports of modules ($\forall t \in (T_{IP} \cup T_{OP})$) are also members of their respective modules. Hence, they are accessible in their respective MOD_PRE and MOD_POST files too. e.g., input port tAI1 is accessible in COMMON_PRE and COMMON_POST (global visibility). As the input port of Alfa, tAI1 is accessible in MOD_Alfa_PRE and MOD_Alfa_POST too (modular visibility).

5. Every transition in a modular Petri net ($\forall t \in T$) has its private visibility. Any transition, be a local member, input or output port of a module, or a member of an inter-modular connector, can have its own processor files. For example, tG1, tAI1, tBO1, tBL2 are are accessible in their own processor files, such as tG1_pre & tG1_post, tAI1_pre & tAI1_post, tBO1_pre & tOB2_post, and tBL2_pre & tBL2_post, respectively, if these files exist.

## 6. The Design of Modules

The important goals of the new design:

- Data hiding: Data hiding inside modules is to abstract away the internal details at higher levels.
- Independent modules: The modules are independent of each other, and have the potential to become autonomous.
- Synchronization of modules: Synchronization ensures that modules must be able to run on different processors. e.g., modules are wrapped as agents and run in parallel in a swarm environment.

Because of these goals (data hiding, independence, parallel execution), *fusion places* and *fusion transitions* are not supported in the new design. Fusion places cannot be allowed, as fusion places allow places in different modules to share internal information. Fusion transitions are not needed either, as in the new design, synchronization is realized at the input or output ports of modules or inside the inter-modular connectors (outside the modules).

### 6.1. Petri Module

In the new design, as shown in Figure 5, modular Petri net model in GPenSIM consists of zero or more Petri Modules. The Petri modules are self-contained and can be developed in isolation and independently tested. The inter-modular connector (IMC, for short) is to connect the modules together.

A Petri module has four distinct sets of elements:

1. **Input ports** $T_{IP}$: Input port transitions function as the input gates of a module. Only through these transitions (input ports), tokens can be directed into the module. These input port transitions have global visibility (accessible in COMMON_PRE and COMMON_POST). Also, due to the belonging to a module, these transitions have local visibility too (accessible in their own modular MOD_PRE and MOD_POST).

2. **Output ports** $T_{OP}$: Output port transitions function as the output gates of a module. Only through these transitions (output ports), tokens can be directed away from the module. Just like the input port transitions, these output port transitions also have global visibility (can be accessed in COMMON_PRE and COMMON_POST), and local visibility too (accessible in the modular MOD_PRE and MOD_POST).

3. **Local transitions** $T_L$: As the local member (internal element) of a module, a local transition consumes tokens from local input places and deposits tokens into local output places. A local transition cannot have any direct connection with the external places (places outside the modules). The local transitions of a module have limited visibility (only modular visibility) as these can be accessed only in the modular MOD_PRE and MOD_POST. The local transitions are not accessible in the global COMMON_PRE and COMMON_POST processors.

4. **Local places** $P_L$: As the local member of a module, a local place feeds tokens to either local transitions or input and output ports of the module. A local place gets tokens from either local transitions or input and output ports of the module. A local place cannot have any direct connection with the external transitions.

### 6.2. Inter-Modular Connectors

In the new design, as shown in Figure 5, a modular Petri net model in GPenSIM consists of zero or more Inter-Modular Connectors (IMC, for short). The IMCs are not modules thus don't possess the input and output ports. They possess IM transitions and IM places:

- **IM transitions** $T_{IM}$: IM transitions have global visibility and are accessible in the COMMON_PRE and COMMON_POST processors. Since IMCs are not modules, these transitions don't have the modular processors MOD_PRE and MOD_POST.
- **IM places** $P_{IM}$: just like the local places inside the modules, the IM places are passive too.

## 7. Formal Definitions for the New Entities

This section presents the formal definitions for the newly designed entities. Let a Place-Transition Petri Net PTN be defined as a four-tuple: $PTN = (P, T, A, M_0)$.

### 7.1. Formal Definition of Petri Module

A Petri Module is defined as a six-tuple:

$$\Phi = (P_{L\Phi}, T_{IP\Phi}, T_{L\Phi}, T_{OP\Phi}, A_\Phi, M_{\Phi 0}),$$

where,

- $T_{IP\Phi} \subseteq T$: $T_{IP\Phi}$ is known as the input ports of the module.
- $T_{L\Phi} \subseteq T$: $T_{L\Phi}$ is known as the local transitions of the module.
- $T_{OP\Phi} \subseteq T$: $T_{OP\Phi}$ is known as the output ports of the module.
- $T_{IP\Phi}$, $T_{L\Phi}$, and $T_{OP\Phi}$, are all mutually exclusive:
  $T_{IP\Phi} \cap T_{L\Phi} = T_{L\Phi} \cap T_{OP\Phi} = T_{OP\Phi} \cap T_{IP\Phi} = \varnothing$.
- $T_\Phi = T_{IP\Phi} \cup T_{L\Phi} \cup T_{OP\Phi}$ (the transitions of the module).
- $P_{L\Phi} \subseteq P$ is known as the set of local places of the module. Since a module has only local places, $P_\Phi \equiv P_{L\Phi}$.
- $\forall p \in P_{L\Phi}$,

  - $\bullet p \in (T_\Phi \cap \varnothing)$. (input transitions of local places are either the transitions of the module or none)
  - $p \bullet \in (T_\Phi \cap \varnothing)$. (output transitions of local places are either the transitions of the module or none)
    This means, local places cannot have direct connections with external transitions.

- $\forall t \in T_{L\Phi}$,

  - $\bullet t \in (P_{L\Phi} \cap \varnothing)$. (input places of local transitions are either the local places or none (cold start))
  - $t \bullet \in (P_{L\Phi} \cap \varnothing)$. (output places of local transitions either the local places or none (sink))

- $\forall t \in T_{IP\Phi}$

  - $\bullet t \in (P_{L\Phi} \cup P_{IM} \cup \varnothing)$. (input places of input ports can be local places or places in inter-modular connectors or can be even an empty set)
  - $t \bullet \in (P_{L\Phi} \cup \varnothing)$. (output places of input ports can only be local places, or empty set)

- $\forall t \in T_{OP\Phi}$

  - $\bullet t \in (P_{L\Phi} \cup \varnothing)$. (input places of output ports can be local places or an empty set)
  - $t \bullet \in (P_{L\Phi} \cup P_{IM} \cup \varnothing)$. (output places of output ports can be local places or places in inter-modular connectors or empty set.

- $A_\Phi \subseteq (P_L \times T_\Phi) \cup (T_\Phi \times P_L)$: where $a_{ij} \in A_\Phi$ is known as the internal arcs of the module.
- $M_{\Phi 0} = [M(p_L)]$ is the initial markings in the local places.

*7.2. Formal Definition of Inter-Modular Connector*

An Inter-modular Connector (IMC) is defined as a four-tuple:

$$\Psi = (P_\Psi, T_\Psi, A_\Psi, M_{\Psi 0})$$

where,

- $P_\Psi \subseteq P$: $P_\Psi$ is the set of places in the IMC (known as the IM-places). $\forall p \in P_\Psi$,

  - $\bullet p \in (T_{OP} \cap T_\Psi \cap \varnothing)$. (input transitions of IM places are either the output ports of modules, IM transitions of this IMC, or none)
  - $p\bullet \in (T_{IP} \cap T_\Psi \cap \varnothing)$. (output transitions of IM places are either the input ports of modules, IM transitions of this IMC, or none)
  This means, IM places cannot have direct connections with local transitions, or IM transitions of other IMCs.

- $\forall p \in P_\Psi, \ \forall i \ \ p \notin P_{\Phi_i}$ (an IM-place cannot be a local place of any Petri module).
- $T_\Psi \subseteq T$: $T_\Psi$ is the transitions of the IMC (known as the IM-transitions). $\forall t \in T_\Phi$,

  - $\bullet t \in (P_\Psi \cap \varnothing)$. (input places of IM-transitions are either the IM-places of this IMC, or none (cold start))
  - $t\bullet \in (P_\Psi \cap \varnothing)$. (output places of IM-transitions either the IM-places of this IMC, or none (sink))

- $\forall t \in T_\Psi, \ \forall i \ \ t \notin T_{\Phi_i}$ (an IM-transition cannot be a transition of any Petri module).
- $A_\Psi \subseteq (P \times T) \cup (T \times P)$: where $a_{ij} \in A_\Psi$ is known as the connecting arcs of the net.
- $M_{\Psi 0} = [M(p_\Psi)]$ is the initial markings in the IM-places.

*7.3. Formal Definition of Modular Petri Net*

A Modular Petri Net is defined as a two-tuple:

$$MPN = (\mathbb{M}, \mathbb{C})$$

where,

- $\mathbb{M} = \sum_{i=0}^{m} \Phi_i$ (zero or more Petri Modules)
- $\mathbb{C} = \sum_{j=0}^{n} \Psi_j$ (zero or more Inter-Modular Connectors)

## 8. Application Example

In this application example, a modular Petri net model is developed for a system involved in computing a quadratic function (e.g., $f = ax^2 + bx + c$). This example is an extended version of the problem stated in [8].

*8.1. The Problem: Computing a Quadratic Function*

The system possesses three communicating agents such as the *client*, and the two workers such as the *multiplier* and the *adder*.

1. The client provides the job to compute, providing the values of the parameters involved (such as a, b, and c).
2. The multiplier performs multiplications. e.g., for an input $(a, x, x)$, multiplier returns $(a \cdot x^2)$. Similarly, if $(b, x)$ is input, multiplier returns $(b \cdot x)$.

3. The adder computes the arithmetic sums. e.g., for an input $(x, y, z)$, adder returns the sum $(x + y + z)$.

The Figure 6 shows the sequence diagram describing the sequences of messages and acknowledgements between the three agents that are involved in performing the job collaboratively.

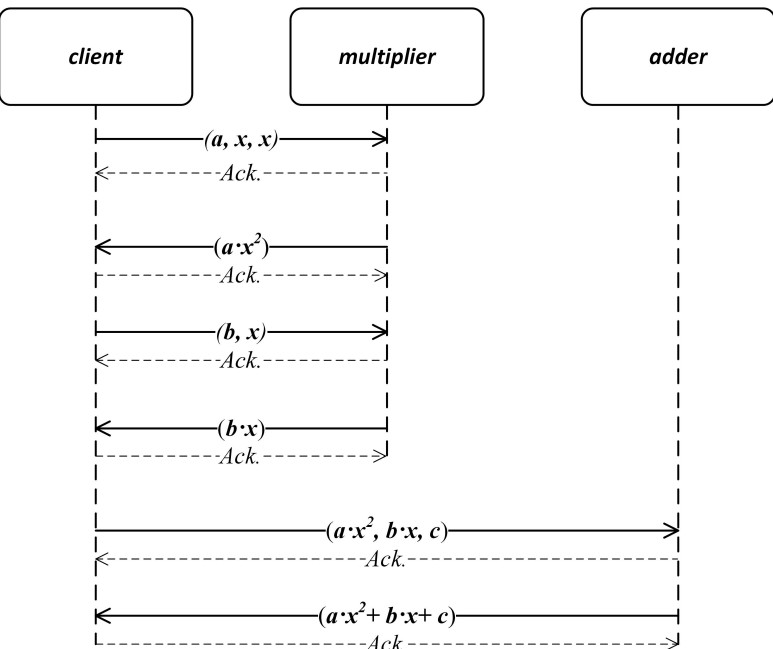

**Figure 6.** The messages and interactions between the agents.

*8.2. Petri Module of a Communicating Agent*

Three main functional entities are usually part of a communicating agent [8]:

1. Observation.
2. Process the inputs and make decisions.
3. Actions.

Thus, the three main functional entities are also represented by some transitions in the Petri module (Figure 7):

- Transition *tCreatMsg* is for creating a message, and *tDispMsg* dispatches the messages. A copy of the transmitted message is kept in the place *pDispdMsg* until the acknowledgement for the message is received.
- Transition *tRecvMsg* receives the messages. Acknowledgement for the received messages is sent by *tAckMsg*. *tProcessMsg* is for processing the arrived message.
- Transition *tRecvAck* is for receiving an acknowledgement for the message that was sent earlier. When an acknowledgement is received then the corresponding message (copy of the message) is removed from the buffer *pDispdMsg*.

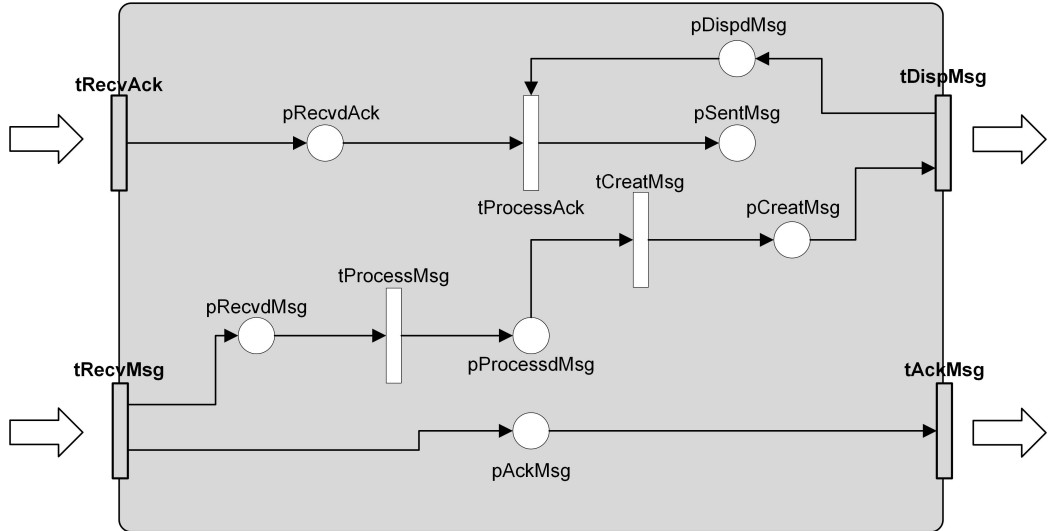

**Figure 7.** Generic Petri module of a communicating agent.

Table 5 explains the functions of the elements of the Petri module for communicating agent.

**Table 5.** Elements of the generic Petri module for communicating agent.

| Element | Purpose |
| --- | --- |
| *tRecvAck* | Receives acknowledgements. |
| *pRecvdAck* | Buffer for received acknowledgements. |
| *tProcessAck* | Processes received acknowledgements. |
| *pSentMsg* | Buffer for storing sent messages. |
| *tCreatMsg* | Creates new messages. |
| *pCreatedMsg* | Buffer for newly created messages |
| *tDispMsg* | Dispatches messages. |
| *pDispdMsg* | Dispatched messages are kept until Ack. are received. |
| *tRecvMsg* | Receives messages. |
| *pRecvdMsg* | Buffer for newly received messages. |
| *tProcessMsg* | Processes received messages. |
| *tProcessdMsg* | Buffer for storing processed messages. |
| *pAckMsg* | Buffer for keeping Ack. before dispatch. |
| *tAckMsg* | Sends acknowledgement for the received messages. |

*8.3. Modular Petri Net Model*

The modular Petri net model is shown in Figure 8. Figure 8 shows that the three agents are represented by Petri modules that are connected via an IMC. All the messages and acknowledgments are passed between the agents in the form of tokens. The data (the values of the parameters in the quadratic function $a$, $b$, $c$, and $x$), the computed values, and the acknowledgements are attached to the tokens as colors. Thus, a Colored Petri net is the backbone of the model.

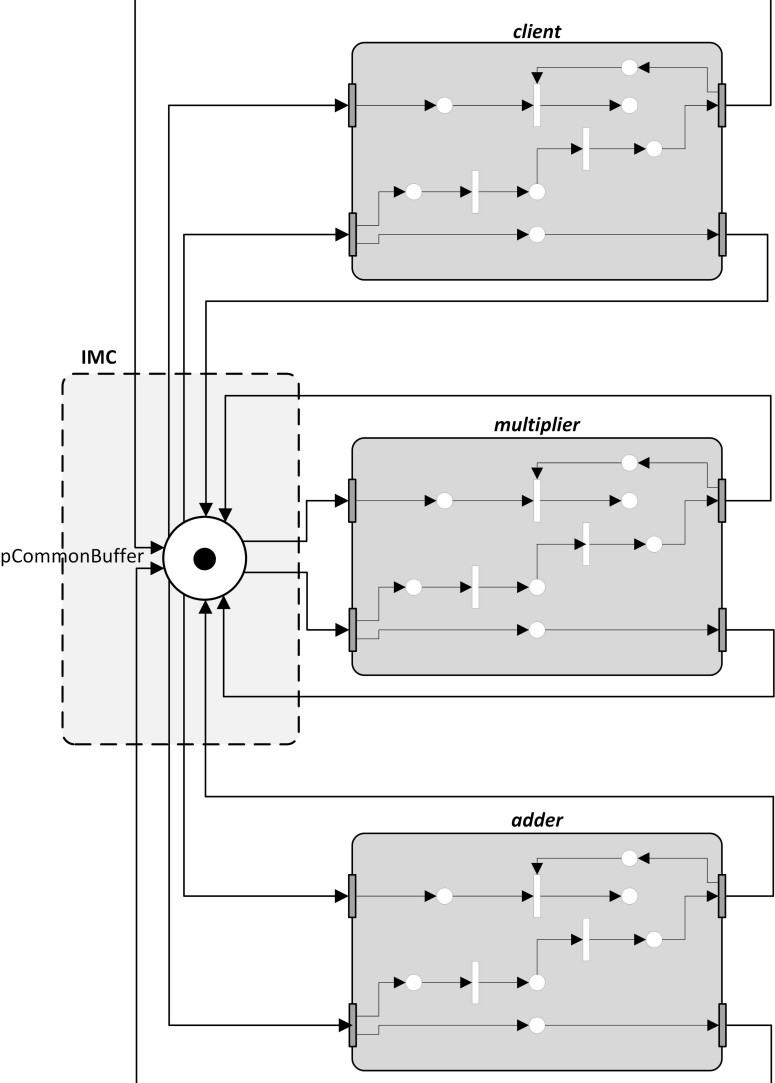

**Figure 8.** The modular Petri model.

## 9. Discussion

This paper presents a new modular Petri net that is designed especially for modeling discrete-event systems that result in large Petri net models. With this modular Petri net, a large model can be decomposed into modules, and these modules can be developed and analyzed independently.

The design approach presented in this paper advocates the use of transitions as the input and output ports of modules. The use of transitions as input and output ports provides the following benefits:

- Active push/pull: Transitions functioning as the input ports can actively pull the tokens from the outside buffers. Hence, these tokens need not be inserted into the modules, which violates data hiding. Also, transitions actively pulling the tokens (e.g., messages) into the modules from the common buffers is a vital mechanism without which modeling intelligent agent will be impossible. Intelligent agents are supposed to act autonomously. In other words, transitions as input and output ports of modules enable modeling independent agents in a peer-to-peer topology.
  In a similar line, transitions functioning as output ports can flush the output tokens of the modules into the output buffers. This property is also important for modeling independent and autonomous modules (e.g., intelligent agents).

- Compression of model: As discussed in the section on literature study, Savi & Xie (1992) [13] presents a powerful technique for module compression, only if the module possesses transitions as input and out ports. However, this technique also demands that the module is modeled as an event-graph.

The uniqueness of this paper is that not only the theory but also the implementation is presented. The theory behind the new modular Petri net presented in this paper is fully implemented in the software GPenSIM. The author of this paper developed GPenSIM. GPenSIM can be downloaded from the website [44]. Even though GPenSIM is a new simulator, it is being used by some universities around the world [45–51]. With the facilities for developing modular Petri net models, it has now become possible to use GPenSIM to model and analyze real-life industrial systems.

This paper is free from any analysis concerning structure and behavior. This is because of brevity; adding the structural and behavioral analysis of the modular Petri net will make this paper significantly large. It may even dilute the rich information presented in this paper on the specifications and definitions. Therefore, a follow-up paper is work-in-progress, which exclusively presents the structural and behavioral analysis of the modular Petri net.

**Limitations of this work**: This paper is the first publication which explains the design of the new Modular Petri net. However, the modular Petri was put to the test on many occasions before; the new Modular Petri net for used for testing large-scale industrial systems, such as airport capacity management [52], modeling oil-drilling activities [53], modeling elevator operations [54], and multi-scale modelling [55]. All these industrial systems were large, and we believe that only with the new modular Petri net, we were able to model, simulate, and performance analyze these systems. The following limitation was experienced during the simulations of large industrial systems: larger the number of modules, slower the simulations will be. Other than simulation time, there were no limitations observed in terms of the number of places, transitions, or tokens.

**Further Work**: We identify two issues as further work: (1) distributed modules, and (2) Control modules.

Distributed modules: One of the goals of developing the new modular Petri net is that the modules must be capable of running on different processors (CPUs). This property will enable modeling the cyber-physical systems that possess components that are geographically separated yet integrated by inter-modular communication. Developing a modular Petri net that enables running modules on different processors is proposed as the further work of this paper. Control modules: The new modular Petri net is not designed with any specific applications in mind. It was designed for generic applications, and to be implemented in the GPenSIM software. However, one of the future goals is to apply the modular Petri net in supervisory control applications. For example, already established supervisory control techniques (e.g., [56,57]) can be wrapped as control modules. Hence, the work towards merging modules with supervisory control techniques is one of the further work of this paper.

**Funding:** This research received no external funding.

**Conflicts of Interest:** The author declares no conflict of interest.

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
