# Peer review of "A New Modular Petri Net for Modeling Large Discrete-Event Systems: A Proposal Based on the Literature Study"

_computers, doi:10.3390/computers8040083_

Round 1
Reviewer 1 Report
This paper proposes a new modular Petri net as a solution to overcome the exponential size of state-spaces modeling the real-life systems. A numerical analysis and comparison study are not provided.
The paper is well written in general and can be considered for publication in “Computers journal” after addressing six main concerns:
The Introduction section is too long. Therefore, I suggest the authors to transfer the section 1.1 into the end of literature section. The authors should widen the related works with slicing of Petri nets by using both time petri nets and the binary decision diagrams. (e.g. such as in http://www.tandfonline.com/doi/abs/10.1080/00207543.2013.848306 or https://sic.ici.ro/volume25-issue1-2016/kammoun/ ). It is better to provide the main limitations of the proposed approach, which used a new modular Petri net. In particular, from how many ‘places, transitions or tokens’ will the proposed algorithm be inefficient? The authors should be propose an optimization methods (exact, heuristic or metaheuristic ) to solve this hard problem In order to show the effectiveness of the proposed approach, it is better to provide comparison results with existing approaches. Finally, as usual, a final thorough proof-reading is recommended.
Author Response
Comments from the Reviewer-1:
This paper proposes a new modular Petri net as a solution to overcome the exponential size of state-spaces modeling the real-life systems. A numerical analysis and comparison study are not provided.
The paper is well written in general and can be considered for publication in “Computers journal” after addressing six main concerns:
Reviewer-1: The Introduction section is too long. Therefore, I suggest the authors to transfer the section 1.1 into the end of literature section.
Author’s response: Section 1.1 is moved to section-2. Thank you for the suggestion!
Reviewer-1: The authors should widen the related works with slicing of Petri nets by using both time petri nets and the binary decision diagrams. (e.g., such as in http://www.tandfonline.com/doi/abs/10.1080/00207543.2013.848306 or
https://sic.ici.ro/volume25-issue1-2016/kammoun/).
Author’s response: We carefully read the two suggested papers; these two papers are on “supervisory control” and management.
The new modular Petri net that is defined in our paper is for general purposes, not specifically for supervisory control or management applications. This is the reason for not mentioning or slanting towards supervisory control.
However, one of the further work (and goal) of the newly designed modular Petri net is to apply it in supervisory control applications.
Hence, the following paragraph is added to the discussion. Thank you for the suggestion!
“The new modular Petri net is not designed with any specific applications in mind. It was designed for generic applications, and to be implemented in the GPenSIM software. However, one of the future goals is to apply the modular Petri net in supervisory control applications. For example, already established supervisory control techniques (e.g., [Kammoun et al, 2014] and [Kammoun et al, 2016]) can be wrapped as control modules. Hence, the work towards merging modules with supervisory control techniques is one of the further work of this paper.”
Reviewer-1: It is better to provide the main limitations of the proposed approach, which used a new modular Petri net. In particular, from how many ‘places, transitions or tokens’ will the proposed algorithm be inefficient?
Author’s response: This is the first time, a paper (this manuscript) on the design of the new modular Petri net is submitted for publication, which explains the design details of the modules. However, the modular Petri was put to the test on many occasions before, for testing large-scale industrial systems, such as airport capacity management, modeling elevator operations, and modeling oil-drilling activities. All these industrial systems were large, and we believe that only with the new modular Petri net, we were able to model, simulate, and performance analyze the system. The only problem (which is also obvious) we encountered was that larger the number of modules, slower the simulations will be. However, apart from the simulation time, we experienced no limitations in terms of the number of places, transitions, or tokens. Hence, the following paragraph is added to the section on the discussion. Thank you very much for your suggestion!
“This paper is the first publication which explains the design of the new Modular Petri net. However, the modular Petri was put to the test on many occasions before; the new Modular Petri net for used for testing large-scale industrial systems, such as airport capacity management [Davidrajuh & Lin, 2015], modeling oil-drilling activities [Davidrajuh & Nejm 2018], and modeling elevator operations [Davidrajuh 2019]. All these industrial systems were large, and we believe that only with the new modular Petri net, we were able to model, simulate, and performance analyze these systems. The following limitation was experienced during the simulations of large industrial systems: larger the number of modules, slower the simulations will be. Other than simulation time, there were no limitations observed in terms of the number of places, transitions, or tokens.”
Reviewer-1: The authors should be propose an optimization methods (exact, heuristic or metaheuristic) to solve this hard problem In order to show the effectiveness of the proposed approach, it is better to provide comparison results with existing approaches.
Author’s response: We regret that we are unable to understand and follow this suggestion. We cannot picture how we can include optimization methods in our paper, under the context of modular Petri net. This paper is about conceiving a new modular Petri net, implement it in a software known as GPenSIM, and then show a sample application. We fervently hope that the reviewer can provide some more help (or hints).
Reviewer-1: Finally, as usual, a final thorough proof-reading is recommended.
Author’s response: We are very sorry that there were some grammatical mistakes, and some ambiguous statements. The paper is now proof-read by a native English speaker, who thoroughly revised the paper.
Reviewer 2 Report
The article fomalizes a novel methodology to modularly represent Petri nets.
The motivations of the work are clearly explained as well as the practical relevance of the proposal. In effect, PNs often become particularly complex when representing real systems and it is possible to incurr in a loss of readability of the graphical model and in the state explosion issue. Moreover, the article is well organized and well written.
My overall opinion on the work is positive, nevertheless some aspects can be improved.
The discussion of the state of the art can be made more comprehensive by including the discussion and comparison with some further works on the modular representation of Petri nets, such as:
10.1109/TASE.2015.2404438
10.1109/TASE.2013.2253552
10.1504/IJSPM.2017.089635
10.1007/s10696-017-9283-9
The formal definition of the technique is consistent. It is not really clear if there is some kind of criterion to build the inte-modular connection and how it can impact on the structural and behavioral analysis of the model. A more detailed discussion should be provided on this aspect.
The case study is really simple and it is not possible to appreciate the capability of the technique to easily and clearly represent large and complex real systems. I suggest to consider a more complex real system and to compare the representation using traditional PNs and the novel methodology.
Author Response
Comments from the Reviewer-2:
The article formalizes a novel methodology to modularly represent Petri nets.
The motivations of the work are clearly explained as well as the practical relevance of the proposal. In effect, PNs often become particularly complex when representing real systems and it is possible to incur in a loss of readability of the graphical model and in the state explosion issue. Moreover, the article is well organized and well written.
My overall opinion on the work is positive, nevertheless some aspects can be improved:
Reviewer-2: The discussion of the state of the art can be made more comprehensive by including the discussion and comparison with some further works on the modular representation of Petri nets, such as: 10.1109/TASE.2015.2404438, 10.1109/TASE.2013.2253552, 10.1504/IJSPM.2017.089635, and 10.1007/s10696-017-9283-9
Author’s response: All the four papers are now included in the literature study. For summary, please check the tables 2 and 3. Thank you for suggesting the papers!
The formal definition of the technique is consistent. It is not really clear if there is some kind of criterion to build the inter-modular connection and how it can impact on the structural and behavioral analysis of the model. A more detailed discussion should be provided on this aspect.
Author’s response:
The suggestion consists of two parts:
Criterion to build inter-modular connectors (IMC). The impact of IMC on the structural and behavioral analysis of the model.For part-1, the following paragraph is added to section-5:
“When a modular model is developed, it happens that there exist one or more elements that cannot be included in any of the modules. The reason can be that the model logic of the modules excludes the inclusion, or simply, the element is an inter-module connector.” “For simplicity, these “leftover” elements can be grouped into a segment (or segments) and be called an IMC.”
For part-2, the following paragraph is added to the discussion:
“This paper is free from any analysis concerning structure and behavior. This is because of brevity; adding the structural and behavioral analysis of the modular Petri net will make this paper significantly large. It may even dilute the rich information presented in this paper on the specifications and definitions. Therefore, a follow-up paper is work-in-progress, which exclusively presents the structural and behavioral analysis of the modular Petri net.”
The case study is really simple and it is not possible to appreciate the capability of the technique to easily and clearly represent large and complex real systems. I suggest to consider a more complex real system and to compare the representation using traditional PNs and the novel methodology.
Author’s response:
Even though this is the first paper that presents the design details of the new modular Petri net, we have already put the modular Petri to the test on many occasions. The new Modular Petri net was used for testing large-scale industrial systems, such as airport capacity management [Davidrajuh & Lin, 2015], modeling oil-drilling activities [Davidrajuh & Nejm 2018], and modeling elevator operations [Davidrajuh 2019]. All these industrial systems were large, and the resulting modular Petri net models were also large. We found out, from our experience, that it will be impossible to explain all the design details we put in this current paper, along with a very large application example.
Hence, we are hesitating to take a big example as the application example. If the reviewer insists on the inclusion of a bigger example (currently, the reviewer kindly suggests), then we will respect his/her request and replace the current application example with a larger one.

Round 2
Reviewer 1 Report
Accept in present form
Reviewer 2 Report
The previously raised issues were sufficiently discussed by the author and the unclear sections were better described.